# Decomposing Causality and Fairness

**Peter Hill**
Decision Science, JPMorgan Chase

**Francois Buet-Golfouse**
Decision Science, JPMorgan Chase
University College London

## Abstract

It is often informative to decompose key quantities of interest into smaller components, in order to develop a better understanding of the key quantity. In this paper, we focus on causality and fairness, where bias attribution can be particularly useful. We show how quantities can be broken down based on independence, or conditional independence criteria, and show how such a decomposition can be used as a diagnosis tool.

## 1 Introduction

Both *causality* (Pearl, 2009; Peters et al., 2017; Neal, 2015; Schölkopf, 2022) and *fairness* (Chouldechova, 2017; Dwork et al., 2012; Donini et al., 2018) are large research topics in themselves, and are closely connected[1]. In both cases, there are key quantities that are of interest to compute. We show how we can decompose key quantities from each area into similar structures. This gives us a simpler way to calculate the quantity, and allows us to understand how close a model is to respecting certain requirements, as well as diagnosing why the model isn't respecting these requirements.

## 2 Causality Decomposition

**Closed-form adjustments** We want to estimate the *causal impact* of a variable $X$ on an outcome $Y$, in the presence of confounders $S$. The quantity $\mathbb{P}(Y|do(X = x)) - \mathbb{P}(Y|X = x)$ tells us how close the interventional distribution and the conditional distribution are. We obtain two decompositions for this quantity based on the *backdoor* and *frontdoor* criteria (see section A.1 of the Appendix), allowing us to consider a causal relationship using purely statistical tools.

**Proposition 1.** *a) Under the same conditions as the **backdoor criteria**, we have*

$$\mathbb{P}(Y|do(X = x)) - \mathbb{P}(Y|X = x) = \sum_s \mathbb{P}(Y|X = x, S = s)\mathbb{P}(S = s)\left[\underbrace{1 - \frac{\mathbb{P}(X = x, S = s)}{\mathbb{P}(X = x)\mathbb{P}(S = s)}}_{\chi_{x,s}}\right] \quad (1)$$

*b) Under the same conditions as the **frontdoor criterion**, we have*

$$\mathbb{P}(Y|do(X = x)) - \mathbb{P}(Y|X = x) =$$
$$\sum_s \mathbb{P}(S = s|X = x)\left(\sum_{x'}[\mathbb{P}(Y|X = x', S = s) - \mathbb{P}(Y|X = x, S = s)]\mathbb{P}(X = x')\right). \quad (2)$$

*Remark* 1. Proposition 1 identifies closed-form expressions for the difference between the interventional distribution of $X$ on $Y$ and the conditional distribution based on the underlying DAG. This allows us to decompose this quantity into sub-components. There is a connection between the backdoor criterion and independence of $X$ and $S$, through $\chi_{x,s}$: the closer $X$ and $S$ to independence, the closer $\chi_{x,s}$ is to 0. Thus the closer $\mathbb{P}(Y|do(X = x))$ and $\mathbb{P}(Y|X = x)$ are. Similarly, the frontdoor criterion is directly connected to the conditional independence of $Y$ and $X$, given $S$.

**Control Variates** These decompositions naturally lead to the idea of control variates (Glasserman & Xu, 2014), where we can make use of an accurate existing estimate, or known value, of $\mathbb{P}(Y|X = x)$, to help model $\mathbb{P}(Y|do(X = x))$, by instead modeling the error from the decomposition.

---

[1]Further details, proofs, and experiments on causality and fairness are provided in the Appendix.

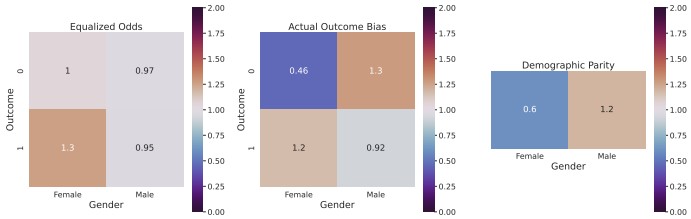

Figure 1: A heatmap showing a decomposition of the EOd-DP formula (Proposition 2) based on the Adult Income dataset. We compare $\text{fr}_{DP}$ with $\text{fr}_{EOd}$ and $\text{fr}_{act}$ for $\hat{y} = 0$. One observes that are close to satisfying the equalised odds criterion (as these values are close to 1), but do not satisfy demographic parity.

## 3  FAIRNESS DECOMPOSITION

We consider $S \in \mathcal{S}$ to be a binary protected attribute, $X \in \mathcal{X}$ to be a set of features (excluding $S$), $Y \in \mathcal{Y}$ to be a binary outcome variable, and $\hat{Y} \in \mathcal{Y}$ to be a predictor for $Y$. Narayanan (2018); Verma & Rubin (2018); Berk et al. (2018); Kim et al. (2020) have reviewed many of the different definitions of fairness that exist. These different definitions cannot all co-exist, as per the impossibility theorems (Kleinberg et al., 2017; Chouldechova, 2017).

| Fairness metric | Equality requirement | Reference |
|---|---|---|
| Demographic parity | $\mathbb{P}(\hat{Y} = \hat{y}\|S = s) = \mathbb{P}(\hat{Y} = \hat{y})$, for all $s \in \mathcal{S}, \hat{y} \in \mathcal{Y}$ | (Calders & Verwer, 2010) |
| Equalized odds | $\mathbb{P}(\hat{Y} = \hat{y}\|, Y = y, S = s) = \mathbb{P}(\hat{Y} = \hat{y}\|Y = y)$, for all $s \in \mathcal{S}, \hat{y}, y \in \mathcal{Y}$ | (Hardt et al., 2016) |

Table 1: Fairness metrics used in this article

**Fairness ratios**  We introduce "fairness ratios" as measures of the deviation of a model from the "fair" situation in terms of ratios, according to a specified fairness metric, Table 1. For example, for Demographic parity, we consider $\text{fr}_{DP}(\hat{y}, s) = \frac{\mathbb{P}(\hat{y}\|s)}{\mathbb{P}(\hat{y})}$, whilst for Equalised odds, we consider $\text{fr}_{EOd}(\hat{y}, s, y) = \frac{\mathbb{P}(\hat{y}\|s, y)}{\mathbb{P}(\hat{y}\|y)}$. We can also consider the actual outcome bias, $\text{fr}_{act}(s, y) = \frac{\mathbb{P}(s\|y)}{\mathbb{P}(s)}$ for all $y \in \mathcal{Y}, s \in \mathcal{S}$. One of the interesting properties of fairness ratios is that they can be decomposed into finer fairness ratios and help diagnose where bias in a model may be coming from.

**Proposition 2.** *EOd-DP Formula Let $y, \hat{y} \in \mathcal{Y}$ and $s \in \mathcal{S}$, then the following equality holds:*

$$fr_{DP}(\hat{y}, s) = \sum_{y \in \mathcal{Y}} fr_{EOd}(\hat{y}, s, y) \cdot fr_{act}(s, y) \cdot \mathbb{P}(y|\hat{y}). \tag{3}$$

*Thus, we can decompose demographic parity in terms of equalized odds and actual bias.*

**Experimentation**  We consider the Adult Income dataset (Dua & Graff, 2017), and have demonstrated the usefulness of this decomposition. We consider the protected attribute, $S$, of gender, and use a simple logistic regression model, without using $S$, to make predictions $\hat{y}$ on if an individual has an income of $> 50K$. We focus on the EOd-DP formula, Proposition 2, and are able to diagnose where DP unfairness is coming from (see Figure 1). To obtain a fair model with respect to DP, one must either tweak the model to be fair w.r.t EO, a form of in-processing, or reweigh the underlying distribution, a form of pre-processing, to avoid outcome disparity.

## 4  CONCLUSION

In this paper, we create decompositions of key interest quantities in causality and fairness and provide closed-form expressions for these decompositions. There are clear similarities that can be drawn between the format of the decompositions and what they allow us to diagnose the source of the impact in each case, providing us with control variate approaches.

## 5 URM STATEMENT

Author Peter Hill meets the URM criteria of the ICLR 2023 Tiny Papers Track.

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

# A    APPENDIX

In this Appendix, we consider the following:

- Additional details on causality (Section A.1
- Additional details on fairness (Section A.3)

## A.1    ADDITIONAL DETAILS ON CAUSALITY

In this section, we provide additional details, proofs and experimental results in relation to causality.

### A.1.1    INTRODUCTORY DETAILS

We typically want to estimate the *causal impact* of a variable $X$ on an outcome $Y$. For example, this could be the causal impact of a treatment on a patient, or a marketing campaign on a KPI. Thorough introductions to the topic of causality can be found in Pearl (2009); Peters et al. (2017); Neal (2015); Schölkopf (2022).

We are able to condition of variable $X$, and deduce the conditional probability $\mathbb{P}(Y|X = x)$. However, this is not equivalent to understanding the causal effect of $X$ on $Y$. Association is not the same as causation. This is due to the fact that there could be confounding variables, $S$, which could be common causes of both $X$ and $Y$.

For example, in the case of a binary outcome $Y$, and binary treatment $X$, we are really interested in understanding the difference between the *potential outcome* of $Y$ under $X$ being 0 or 1. However, since only one of these outcomes can occur (either the treatment can be given or not), we cannot observe both outcomes. This is the Fundamental Problem of Causal Inference.

In order to remove the effect of the confounders, we must instead intervene on $X$, which we write as $do(X = x)$. Intervening allows us set the value of $X$, thus removing the effect of any confounding variables on $X$. We are then interested in $\mathbb{P}(Y|do(X = x))$, which is a *causal estimand*. For example, in the case of a binary variable $X \in 0, 1$ (such as being given a treatment or not), in order to establish the causal effect of $X$ on $Y$, we must consider $\mathbb{E}[Y|do(X = 1)] - \mathbb{E}[Y|do(X = 0)]$.

In order to understand the causal impact, we must first introduce the idea of a Directed Acyclic Graph (DAG), which assumes the causal structure of the problem. For example, we could consider the causal structures outlined in Figures 2 or 3a. Key structures of causal graphs are *chains*, *forks*, and *colliders*, which are outlined in Figure 2. In these cases, we refer to $S$ as the middle node.

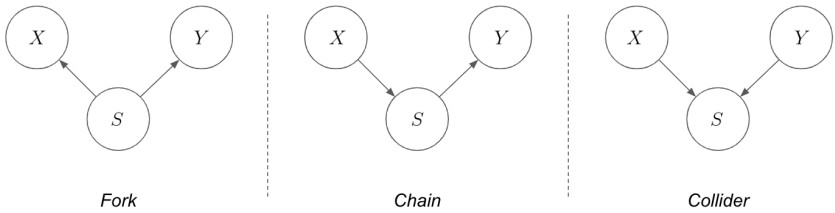

Figure 2: Forks, Chains and Colliders

Key results in causal inference theory allow us to relate these interventional distributions, which we are most interested in, to conditional distributions. This allows us to consider a causal relationship using purely statistical tools. We consider these results, in particular the backdoor and frontdoor criteria discussed in the following.

**Definition 1.** A *backdoor path* from $X$ to $Y$ is a non-causal path from $X$ to $Y$.

**Definition 2.** A path between $X$ and $Y$ is blocked by a node $S$ if either

- There is a fork, or a chain, on the path, with $S$ as the middle node, where $S$ is conditioned on.

- There is a collider on the path, with $S$ as the middle node, where neither $S$, nor any descendants of $S$ are conditioned on.

**Theorem 1** (Backdoor Criterion). *Suppose that a set $S$ blocks all backdoor paths (non-causal paths) from $X$ to $Y$, and does not contain any descendants of $X$, then*

$$\mathbb{P}(Y|do(X=x)) = \sum_{s \in \mathcal{S}} \mathbb{P}(Y|X=x, S=s)P(S=s) \tag{4}$$

**Theorem 2** (Frontdoor Criterion). *Suppose that a set $S$ completely mediates the effect of $X$ on $Y$, that there is no unblocked backdoor path from $X$ to $S$, and that all backdoor paths from $S$ to $Y$ are blocked by $X$, then*

$$\mathbb{P}(Y|do(X=x)) = \sum_{s} \mathbb{P}(S=s|X=x) \sum_{x'} \mathbb{P}(Y|X=x', S=s)P(X=x'). \tag{5}$$

### A.1.2 PROOFS

**Proposition 1.** *a) Under the same conditions as the **backdoor criteria**, we have*

$$\mathbb{P}(Y|do(X=x)) - \mathbb{P}(Y|X=x) = \sum_{s} \mathbb{P}(Y|X=x, S=s)\mathbb{P}(S=s)\left[1 - \underbrace{\frac{\mathbb{P}(X=x, S=s)}{\mathbb{P}(X=x)\mathbb{P}(S=s)}}_{\chi_{x,s}}\right] \tag{1}$$

*b) Under the same conditions as the **frontdoor criterion**, we have*

$$\mathbb{P}(Y|do(X=x)) - \mathbb{P}(Y|X=x) =$$

$$\sum_{s} \mathbb{P}(S=s|X=x)\left(\sum_{x'}[\mathbb{P}(Y|X=x', S=s) - \mathbb{P}(Y|X=x, S=s)]\mathbb{P}(X=x')\right). \tag{2}$$

*Proof of a).* Firstly, we recall that using the backdoor criteria, we have

$$\mathbb{P}(Y|do(X=x)) = \sum_{s \in \mathcal{S}} \mathbb{P}(Y|X=x, S=s)P(S=s). \tag{6}$$

Also, we note that

$$\mathbb{P}(Y|X=x) = \sum_{s \in \mathcal{S}} \mathbb{P}(Y|X=x, S=s)\mathbb{P}(S=s|X=x). \tag{7}$$

Combining these together, we get that

$$\mathbb{P}(Y|do(X=x)) - \mathbb{P}(Y|X=x)$$

$$= \sum_{s} \bigg( \mathbb{P}(Y|X=x, S=s)\mathbb{P}(S=s) - \mathbb{P}(Y|X=x, S=s)\mathbb{P}(S=s|X=x) \bigg)$$

$$= \sum_{s} \mathbb{P}(Y|X=x, S=s)\bigg( \mathbb{P}(S=s) - \mathbb{P}(S=s|X=x) \bigg)$$

$$= \sum_{s} \mathbb{P}(Y|X=x, S=s)\mathbb{P}(S=s)\chi_{x,s} \tag{8}$$

where the final line comes from the fact that

$$\mathbb{P}(S=s|X=x) = \frac{\mathbb{P}(S=s, X=x)}{\mathbb{P}(X=x)} \tag{9}$$

$\square$

*Proof of b).* From the frontdoor criterion, we note that

$$\mathbb{P}(Y|do(X=x)) = \sum_{s} \mathbb{P}(S=s|X=x) \sum_{x'} \mathbb{P}(Y|X=x', S=s)P(X=x'). \tag{10}$$

Also, we note that

$$\mathbb{P}(Y|X=x) = \sum_s \mathbb{P}(Y|X=x, S=s)\mathbb{P}(S=s|X=x). \tag{11}$$

Combining these together we see that

$$\mathbb{P}(Y|do(X=x)) - \mathbb{P}(Y|X=x)$$

$$= \sum_s \left( \mathbb{P}(S=s|X=x)\left(\sum_{x'}\mathbb{P}(Y|X=x', S=s)P(X=x')\right) - \mathbb{P}(Y|X=x, S=s)\mathbb{P}(S=s|X=x)\right)$$

$$= \sum_s \mathbb{P}(S=s|X=x)\left(\left(\sum_{x'}\mathbb{P}(Y|X=x', S=s)P(X=x')\right) - \mathbb{P}(Y|X=x, S=s)\right) \tag{12}$$

Further, note that

$$\mathbb{P}(Y|X=x, S=s) = \mathbb{P}(Y|X=x, S=s)\sum_{x'}\mathbb{P}(X=x') \tag{13}$$

$$= \sum_{x'}\mathbb{P}(Y|X=x, S=s)\mathbb{P}(X=x') \tag{14}$$

Combining the above, we see that

$$\mathbb{P}(Y|do(X=x)) - \mathbb{P}(Y|X=x) =$$

$$\sum_s \mathbb{P}(S=s|X=x)\left(\sum_{x'}\mathbb{P}(Y|X=x', S=s)P(X=x') - \sum_{x'}\mathbb{P}(Y|X=x, S=s)\mathbb{P}(X=x')\right)$$

$$= \sum_s \mathbb{P}(S=s|X=x)\left(\sum_{x'}[\mathbb{P}(Y|X=x', S=s) - \mathbb{P}(Y|X=x, S=s)]\mathbb{P}(X=x')\right) \tag{15}$$

$\square$

## A.2 Further Results

We can further derive results based on causality ratios, allowing us to draw further similarities to the fairness setting.

**Proposition 3.** *Under the same conditions as the **backdoor criteria**, we have*

$$\frac{\mathbb{P}(Y|do(X=x))}{\mathbb{P}(Y|X=x)} = \sum_s \frac{\mathbb{P}(S=s|Y, X=x)}{\mathbb{P}(S=s|X=x)}\mathbb{P}(S=s) \tag{16}$$

*Proof.* Using the backdoor criterion, we have that

$$\frac{\mathbb{P}(Y|do(X=x))}{\mathbb{P}(Y|X=x)} = \sum_s \mathbb{P}(Y|X=x, S=s)\mathbb{P}(S=s). \tag{17}$$

Bayes Rule gives that

$$\mathbb{P}(Y|X=x, S=s) = \frac{\mathbb{P}(S=s|Y, X=x)\mathbb{P}(Y|X=x)}{\mathbb{P}(S=s|X=x)}. \tag{18}$$

Substituting Equation 18 into Equation 17, and cancelling the $\mathbb{P}(Y|X=x)$ term, we get the result.

$\square$

### A.2.1 Control Variates

If we wish to model $\mathbb{P}(Y|do(X=x))$, these decompositions naturally lead to the idea of control variates. The aim is to make use of existing solutions to similar problems, to solve a more difficult problem. Supposing that the conditional probability $\mathbb{P}(Y|X=x)$ was known, a control variates approach would be to model the differences from Proposition 1, and add it on to $\mathbb{P}(Y|X=x)$, rather than modeling $\mathbb{P}(Y|do(X=x))$ directly.

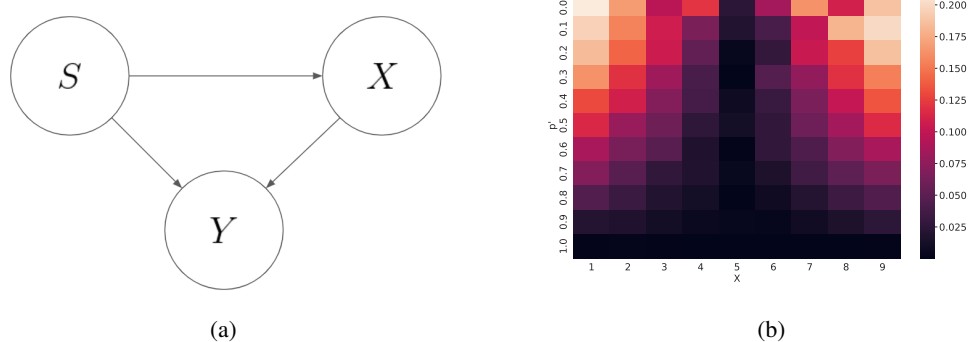

(a)                                                    (b)

Figure 3: a) DAG used in the synthetic dataset. b) Absolute value of quantity from Proposition 1 a) for synthetic data, for different values of $p'$. A darker colour gives a value closer to $0$, thus indicating less dependence between $X$ and $S$. For each $x$ as we decrease $p'$ from $1$ to $0$, we introduce more dependency between $X$ and $S$, and thus our quantity increases in absolute value, and the conditional distribution is further away from the interventional distribution.

### A.2.2 EXPERIMENTATION

We run experiments in relation to the causality decompositions. We consider a synthetic dataset under the DAG represented in Figure 3a. Here, we are interested in understanding the causal relationship of $X$ on $Y$. However, there is also a confounder, $S$. In order to generate the data, we randomly sample $s_i \in [1, \cdots, 10]$, and latent $u_i \in [1, \cdots, 10]$ for $i = 1, \cdots, 100000$. Then we consider $x_i = p'u_i + (1 - p')s_i$, for some $p' \in (0, 1)$, which controls the covariance between $S$ and $X$. We model $Y_i \sim Bernoulli(p_i)$, where $p_i = \frac{x_i + s_i}{\max_i x_i + \max_i s_i}$. Note that $S$ satisfies the backdoor criterion conditions, so we can use Proposition 1 a).

Figure 3b shows the value of the quantity $\mathbb{P}(Y|do(X = x)) - \mathbb{P}(Y|X = x)$, for each $x$ and different values of $p'$. Varying $p'$ from between $0$ and $1$ allows us to vary the covariance between $X$ and $S$. For smaller $p$, we have higher covariance, which means that the quantity $\chi_{x,s}$ is larger, and thus the quantity $\mathbb{P}(Y|do(X = x)) - \mathbb{P}(Y|X = x)$ is also larger.

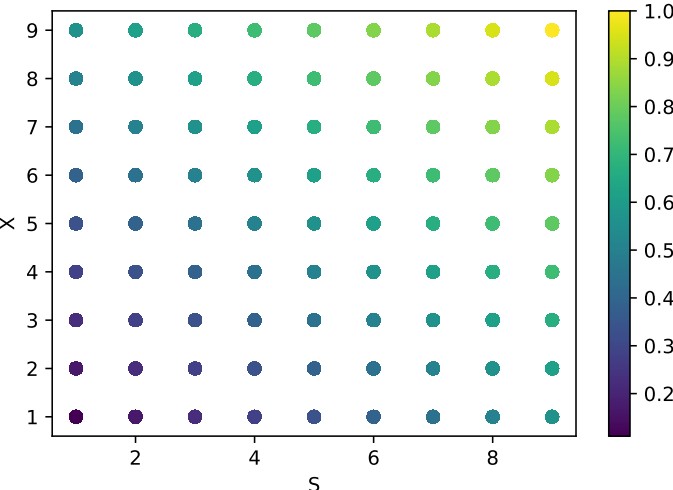

Figure 4: An example of the synthetic data used, with $p' = 0.5$. The colour indicates the value of $p$.

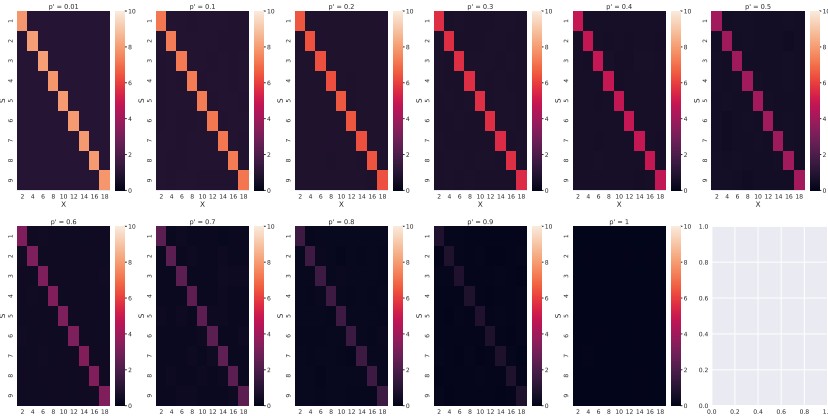

Figure 5: The values of $\chi_{x,s}$ for different values of $p'$ using the synthetic data. We see that the closer $p'$ is to 1, the closer to independence we are, and thus the closer $\chi_{x,s}$ is to 0.

### A.3 ADDITIONAL DETAILS ON FAIRNESS

In this section, we provide additional details, proofs and experimental results in relation to fairness.

#### A.3.1 PROOFS

**Proposition 2.** *EOd-DP Formula* *Let* $y, \widehat{y} \in \mathcal{Y}$ *and* $s \in \mathcal{S}$, *then the following equality holds:*

$$fr_{DP}(\widehat{y}, s) = \sum_{y \in \mathcal{Y}} fr_{EOd}(\widehat{y}, s, y) \cdot fr_{act}(s, y) \cdot \mathbb{P}(y|\widehat{y}). \tag{3}$$

*Thus, we can decompose demographic parity in terms of equalized odds and actual bias.*

*Proof.* Firstly, we note that this is equivalent to proving the following:

$$\frac{\mathbb{P}(\widehat{y}|s)}{\mathbb{P}(\widehat{y})} = \sum_{y \in \mathcal{Y}} \frac{\mathbb{P}(\widehat{y}|s, y)}{\mathbb{P}(\widehat{y}|y)} \cdot \frac{\mathbb{P}(s|y)}{\mathbb{P}(s)} \cdot \mathbb{P}(y|\widehat{y}). \tag{19}$$

We note that

$$\frac{\mathbb{P}(\widehat{y}|s)}{\mathbb{P}(\widehat{y})} = \sum_{y \in \mathcal{Y}} \frac{\mathbb{P}(\widehat{y}, y|s)}{\mathbb{P}(\widehat{y})} \tag{20}$$

$$= \sum_{y \in \mathcal{Y}} \frac{\mathbb{P}(\widehat{y}, |y, s)\mathbb{P}(y|s)}{\mathbb{P}(\widehat{y})} \tag{21}$$

$$= \sum_{y \in \mathcal{Y}} \frac{\mathbb{P}(\widehat{y}, |y, s)\mathbb{P}(s|y)\mathbb{P}(y)}{\mathbb{P}(\widehat{y})\mathbb{P}(s)}, \tag{22}$$

where the second line comes from the definition of conditional probability, and the third line comes from Bayes Theorem. Also, note that

$$\mathbb{P}(\hat{y}|y) = \frac{\mathbb{P}(y, \hat{y})}{\mathbb{P}(y)} \tag{23}$$

and

$$\mathbb{P}(y|\hat{y}) = \frac{\mathbb{P}(y, \hat{y})}{\mathbb{P}(\hat{y})}. \tag{24}$$

Combining, we obtain,

$$\frac{\mathbb{P}(y)}{\mathbb{P}(\hat{y})} = \frac{\mathbb{P}(y|\hat{y})}{\mathbb{P}(\hat{y}|y)}. \tag{25}$$

Replacing this ratio in Equation 22, we obtain the result. □

Using this result, we can decompose demographic parity in terms of equalized odds and actual bias. In particular, if the assumption of equalized odds is verified, then $\mathrm{fr}_{EOd}(\hat{y}, s, y) = 1$ for all $y \in \mathcal{Y}$, then $\mathrm{fr}_{DP}(\hat{y}, s) = \sum_{y \in \mathcal{Y}} \mathrm{fr}_{act}(s, y) \cdot \text{model probability}(y, \hat{y})$. If, in addition, $\mathrm{fr}_{act}(s, y) = 1$, then $\mathrm{fr}_{DP}(\hat{y}, s) = 1$, but if $\mathrm{fr}_{act}(s, y) \neq 1$, we recover the well-known result that demographic parity does not hold in general (see (Kleinberg et al., 2017)).

### A.3.2 EXPERIMENTATION

Figure 6 demonstrates the decomposition of the EOd-DP Formula for the Adult Income dataset, with the set-up outlined in the main text. The top row corresponds to $\hat{y} = 0$, whilst the bottom row corresponds to $\hat{y} = 1$.

Figure 7 demonstrates the decomposition of the EOd-DP Formula (Proposition 2) when we fix $\mathrm{fr}_{EOd} = 1$. It shows us that even if we have a completely fair model with respect to Equalised odds, this does not guarantee that our model will be fair under Demographic parity.

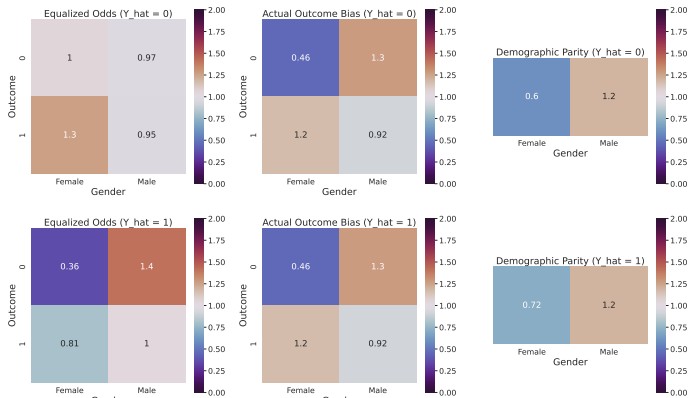

Figure 6: A heatmap showing a decomposition of the EOd-DP formula (Proposition 2) based on the Adult Income dataset. Comparing $\mathrm{fr}_{DP}$ with $\mathrm{fr}_{EOd}$ and $\mathrm{fr}_{act}$. We see that for $\hat{y} = 0$ (the top row), we are close to satisfying the equalised odds criterion, but do not satisfy demographic parity, whilst in the case $\hat{y} = 1$ (the bottom row), we do not generally satisfy either condition.

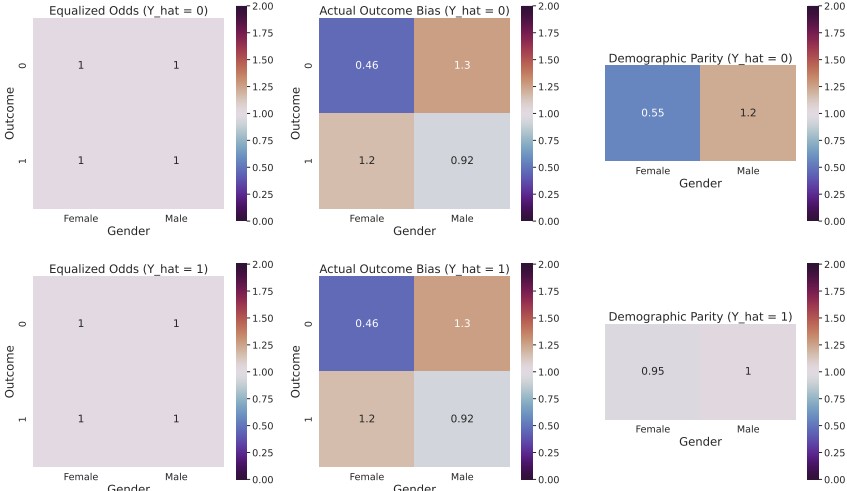

Figure 7: A heatmap showing a decomposition of the EOd-DP formula (Proposition 2) based on the Adult Income dataset, when we fix $\mathrm{fr}_{EOd} = 1$, thus assuming a fair model with respect to Equalised Odds. We see that we aren't guaranteed to obtain fairness with respect to Demographic Parity.

