# OpenReview forum: "Decomposing Causality and Fairness"
_ICLR.cc/2023/TinyPapers — Submitted to Tiny Papers @ ICLR 2023_

### Official Review · Reviewer_sG32 · 2023-03-27

**Confidence:** 2

**Summary Of Contributions:**

In this paper authors investigated key interest quantities in causality and fairness to identify the source of impact.

**Rating:**

High Potential (HP): a submission which meets the reviewing criteria and has potential to make an impact on the field

**Strengths And Weaknesses:**

Strengths:

-The paper is well justified with the propositions

-The authors show the working of the proposed solution with a publicly available dataset

Weakness:

-The authors should explain whether causality decomposition and fairness decomposition are linked together. We suggest them compare the results of individual decompositions with their proposed solution

- We suggest authors run a grammar check throughout the paper and improve it for better understanding; for example, "and allows us to understand the how close a model is to respecting certain requirements" should be "and allows us to understand how close a model is to respecting certain requirements"

**Suggested Changes:**

-The authors should explain whether causality decomposition and fairness decomposition are linked together. We suggest them compare the results of individual decompositions with their proposed solution

- We suggest authors run a grammar check throughout the paper and improve it for better understanding; for example, "and allows us to understand the how close a model is to respecting certain requirements" should be "and allows us to understand how close a model is to respecting certain requirements."

---

### Official Review · Reviewer_iJC4 · 2023-04-02

**Confidence:** 4

**Summary Of Contributions:**

This paper focuses on causality and fairness, where bias attribution can be particularly useful and shows how a decomposition of causality and fairness can be used for fairness.

**Rating:**

High Potential (HP): a submission which meets the reviewing criteria and has potential to make an impact on the field

**Strengths And Weaknesses:**

**Strengths**
1. The relationship between causality and fairness is important for addressing fairness issues. This paper undertakes an important and interesting exploration of this topic.
2. This paper is well-presented and easy to follow.
3. This paper demonstrates a strong understanding of fairness and causality and proposes an effective way to utilize them.


**Weaknesses**\
I don’t see obvious weaknesses in this paper.


**Suggested Changes:**

N/A

---

### Author Response · Authors · 2023-05-31
**Opt-in for archival**

The authors wish to opt-in for archival in the ICLR Tiny Papers track.

---

### Meta-Review · Area_Chair_SQ3u · 2023-04-07

**Recommendation:** Invite to present (notable)
**Confidence:** 4

**Metareview:**

The paper received two reviews, both of which were positive about the paper's potential impact in the field of causality and fairness. The reviewers found the paper to be well-presented and easy to follow, demonstrating a strong understanding of the topic. While one reviewer had some concerns, they still rated the paper as having high potential and appreciated the authors' contributions.

**Summary:**

The paper proposes a decomposition method for causality and fairness that can be used to address fairness issues. The paper is well-justified, and the proposed solution is demonstrated with a publicly available dataset. The paper has high potential to make an impact in the field of causality and fairness with an interesting exploration.

**Comments And Feedback To The Authors:**

The authors should consider addressing the concerns raised by reviewer sG32 regarding the explanation of the causality decomposition and fairness decomposition and how they are linked together. Additionally, they could improve the clarity of the writing by running a grammar check throughout the paper. Overall, we appreciate the authors' contributions to the field of causality and fairness.

**Reason For Not Giving A Higher Recommendation:**

One reviewer expressed some concerns about the explanation of causality decomposition and fairness decomposition and how they are linked together. We suggest them compare the results of individual decompositions with their proposed solution

**Reason For Not Giving A Lower Recommendation:**

The paper addresses an important and interesting topic, the relationship between causality and fairness, which is crucial for addressing fairness issues.
The paper is well-justified and proposes an effective way to utilize causality and fairness.
The proposed solution is demonstrated with a publicly available dataset, which adds credibility to the paper's claims.
The paper is well-presented and easy to follow, demonstrating a strong understanding of the topic.

---

### Decision · Program_Chairs · 2023-04-09

Invite to present (notable)